# Exposure to Hantavirus is a Risk Factor Associated with Kidney Diseases in Sri Lanka: A Cross Sectional Study

**DOI:** 10.3390/v11080700

**Published:** 2019-07-31

**Authors:** Yomani D. Sarathkumara, Chandika D. Gamage, Sithumini Lokupathirage, Devinda S. Muthusinghe, Nishantha Nanayakkara, Lishanthe Gunarathne, Kenta Shimizu, Yoshimi Tsuda, Jiro Arikawa, Kumiko Yoshimatsu

**Affiliations:** 1Department of Microbiology, Faculty of Medicine, University of Peradeniya, Peradeniya 20400, Sri Lanka; 2Department of Microbiology and Immunology, Faculty of Medicine, Hokkaido University, Sapporo 060-8638, Japan; 3Graduate School of Infectious Diseases, Hokkaido University, Sapporo 060-8638, Japan; 4Nephrology and Transplantation Unit, Teaching Hospital Kandy, Kandy 20000, Sri Lanka; 5Renal Clinic, District Hospital, Girandurukotte 90750, Sri Lanka

**Keywords:** Thailand orthohantavirus, tropical nephropathy, rodent

## Abstract

Chronic kidney disease of unknown etiology (CKDu) imposes a substantial burden on public health in Sri Lankan agricultural communities. High seroprevalences of hantavirus have been reported in CKDu patients in several locations of Sri Lanka. We carried out a cross-sectional study followed by an unmatched case-control comparison in two geographically distinct areas of Sri Lanka, Girandurukotte (CKDu endemic) and Kandy (CKDu non-endemic) to determine whether exposure to hantaviruses is a potential risk factor in patients with kidney disease. An indirect immunofluorescent antibody assay using two antigens, Thailand orthohantavirus-infected and recombinant N protein-expressing Vero E6 cells, were used for serodiagnosis. Participants’ demographic and other socio-economic data were collected through a structured questionnaire. Fifty kidney disease patients and 270 controls from Kandy and 104 kidney disease patients and 242 controls from Girandurukotte were examined. Seropositivities were 50% and 17.4% in kidney patients and controls, respectively, in Girandurukotte, and they were 18% and 7% in Kandy. The odds of exposure to hantaviruses were higher for kidney disease patients than for controls in both Girandurukotte (OR:3.66, 95% CI:2.01 to 6.64) and Kandy (OR:2.64, 95% CI:1.07 to 6.54) in binary logistic regression models. According to statistical analysis, individuals exposed to hantaviruses had a higher risk of developing renal impairment. Therefore, hantavirus infection might be an important risk factor for development of kidney disease in Sri Lanka.

## 1. Introduction

Chronic kidney disease (CKD) is a major global public health problem [1]. Diabetes mellitus and hypertension are the most common causes of CKD in developed countries. However, a severe form of CKD that has been reported in individuals without any association of these known risk factors is defined as CKD of unknown etiology (CKDu) [2]. CKDu has emerged as a significant public health crisis in rural agricultural communities in Sri Lanka, among sugarcane farmers in Central American countries, and in some areas of India and Egypt [3]. In Sri Lanka, the number of CKDu patients has been increasing rapidly since the 1990s, mainly in the dry zones of North-Central, Uva and North Western provinces. CKDu prevalence ranges from 15.1% to 22.9% in highly prevalent areas [4].

Hantaviruses belong to the order Bunyavirales, family Hantaviridae, and genus Orthohantavirus and they are naturally maintained in rodent and other small mammal reservoirs [5]. Several species of hantaviruses cause two important zoonoses: Hemorrhagic fever with renal syndrome (HFRS) in the Old World and hantavirus pulmonary syndrome (HPS) in the New World [6]. In addition to pathogenic viruses, many hantaviruses are known to have low pathogenicity or no pathogenicity in humans. We recently reported high seroprevalence of hantaviruses among CKDu patients in Girandurukotte, one of the CKDu hotspots in Sri Lanka [7]. Preliminary findings suggested that Thailand orthohantavirus (THAIV) or a THAIV-related virus was circulating in this area [8]. THAIV was originally isolated in Thailand in 1985 [9]. However, THAIV is not associated with classic HFRS in Asia, and its pathogenicity in humans has not been determined yet [10,11,12]. Moreover, the relationship between hantavirus infection and CKDu in Sri Lanka has not been evaluated, as there were no comparative analyses between a CKDu endemic and a non-endemic area in previous studies.

In this study, we conducted a cross-sectional study followed by an unmatched case-control comparison to identify risk factors of exposure to THAIV among renal disease patients and healthy controls in two geographically distinct areas: endemic area (Girandurukotte) and a non-endemic area (Kandy).

## 2. Materials and Methods

### 2.1. Selection of Diagnostic Method for CKDu Patient Sera

We reported that anti-THAIV antibodies were detected from sera collected in Girandurukotte in 2016 and these sera were used to select an appropriate serological diagnostic assay [8]. However, THAIV or a THAIV-related hantavirus has not been isolated in Sri Lanka and thus could not be used as a homologous antigen for serological assays. Recombinant N protein (rNP) antigens expressed by *Escherichia coli* (*E. coli*), baculovirus, and mammalian expression vector (pCAGGS/MCS) were compared in ELISA and an indirect immunofluorescent antibody assay (IFA).

#### 2.1.1. Virus and Cells

The THAIV strain Thai749 was propagated in Vero E6 cells (ATCC C1008) [9]. Vero E6 cells were grown in Eagle’s minimum essential medium (EMEM; Gibco, Thermo Fisher Scientific, Life Technologies, Waltham, MA, USA) supplemented with 5% heat-inactivated fetal bovine serum (Biowest, Nuaille, France), penicillin, streptomycin (Sigma Aldrich CO. St Louis, MO, USA), and ITS supplement (insulin-transferrin-selenium, Gibco). High Five cells were grown in Grace’s medium (Gibco) supplemented with 10% heat-inactivated fetal bovine serum (Biowest), penicillin and streptomycin (Sigma Aldrich CO.). Recombinant baculovirus to express rNP of Seoul virus (SEOV) strain SR-11 was inoculated into High Five cells [13].

#### 2.1.2. Indirect Immunofluorescent Antibody Assay

The prototype IFA antigen was prepared and used as described previously [14]. Briefly, THAIV or mock-infected Vero E6 cells were trypsinized and cultivated on 24-well glass slides (Matsunami Glass, Osaka, Japan) overnight at 37 °C in 5% CO_2_. Cell sheets on the wells were washed with PBS and then fixed with acetone at room temperature for 10 min and rinsed with distilled water. After drying, the slides were stored at −80 °C until further use. Alexa Fluor 488-conjugated protein A (Invitrogen, Thermo Fisher Scientific, Life Technologies) was used as the secondary antibody, and IFA tests were performed at a serum dilution of 1:100. Serum showing a clear granular immunofluorescence pattern in the cell cytoplasm of THAIV-infected Vero E6 cells was detected as positive, and a serum specimen showing a non-specific fluorescent profile in both mock and THAIV infected Vero E6 cell antigens were detected as negative. The judgment of the IFA test results was recorded double-blinded by two to four examiners.

#### 2.1.3. Preparation of an rNP-Based IFA Antigen

A recombinant NP antigen of THAIV was originally prepared for serological analyses of rodent sera [15]. The amplified NP gene tagged with ClaI and XhoI recognition sites was cloned into a mammalian expression plasmid vector, pCAGGS/MCS following ClaI and XhoI restriction treatment of the NP amplicons and pCAGGS/MCS. Vero E6 cells were transfected with pCAGGS/MCS containing the coding region of THAIV NP by TransIT-LT-1 transfection reagent (Mirus Bio, Madison, WI, USA) according to the manufacturer’s instructions. IFA slides containing the rNP-based antigen were prepared as previously described. Transfection efficiency of Vero E6 cells was estimated to be about 5–10% according to the ratio of rNP-expressing cells detected by monoclonal antibodies E5/G6 directed to NP of Hantaan orthohantavirus (HTNV) described in detail below.

#### 2.1.4. Mouse Monoclonal Antibodies and Its IFA Profiles

Mouse monoclonal antibodies directed to HTNV NP, E5/G6 and HTNV envelope glycoproteins (GP), 1G8 antibodies, were used to confirm viral antigens [14,16,17]. Alexa Fluor 488-conjugated goat anti-mouse IgG (Invitrogen) was used as the secondary antibody. Immunofluorescent patterns of NP and GP in both IFA antigens, prototype THAIV, and rNP-based THAIV, were observed.

#### 2.1.5. ELISA by Using Recombinant NPs

ELISA was carried out as previously described [13,18,19]. Briefly, the entire rNP of SEOV was expressed in High Five cells by baculovirus vector, and the cell lysate was used as the antigen. The cell lysate of High Five cells inoculated with baculovirus lacking polyhedrin gene was used as a negative antigen. The ELISA OD value for the rNP antigen was obtained after subtraction of the ELISA OD value for the negative control antigen. A truncated NP antigen of N-terminal 103 amino acids of HTNV named as HS103 antigen was expressed by an *E. coli* vector and applied for serological diagnosis for rodents in ELISA and immunochromatography (ICG) [10,20,21]. Goat anti-human IgG (H+L) labeled with horseradish peroxidase (Kirkegaard and Perry Laboratories, Inc. (KPL), Gaithersburg, MD, USA) was used as a secondary antibody in ELISA with serum diluted at 1:500.

### 2.2. Study Design, Study Population and Sample Size Calculation for the Cross-Sectional Study

This was a hospital- and community-based cross-sectional study followed by an unmatched case-control comparison. The study was conducted in the Kandy and Badulla Districts (Figure 1a). According to the study objectives, a cross-sectional study was performed to obtain demographic information and to collect blood samples for laboratory analyses (Table 1). The determination of sample size is described in Appendix A.

#### 2.2.1. Hospital-Based Study

Renal patients were recruited according to the information on patients’ clinical record books as diagnosed by the Consultant Nephrologist at Nephrology and Transplantation Unit of Kandy Teaching Hospital (KTH) and residential clinician at the renal clinic, Girandurukotte District hospital (GK DH) according to the standard guidelines [22,23]. Patients clinically diagnosed with a kidney disease residing in the Kandy and Badulla districts and who attended the regular renal clinics in both hospitals were recruited for this study. The sample size from GK DH was estimated, with an approximate prevalence of CKDu of 15%, as reported elsewhere [4]. We estimated the sample sizes for GK DH as 196 and for KTH as 100. Patients (18 years and above) who were clinically diagnosed with kidney disease, who resided in one of the two districts, and who attended the renal clinics conducted by KTH or GK DH were included in this study. Patients of age under 18 years were excluded.

#### 2.2.2. Community-Based Study

Medical Office of Health (MOH) areas were randomly selected within the Kandy District. The Regional Director of Health Services for Kandy and Badulla granted permission for collecting blood samples and demographic data from the community in the selected MOH/Public Health Inspector (PHI) areas. People who were 18 years and above, who had no past history or current diagnosis of renal disease, resided in Kandy or Girandurukotte, and who voluntarily provided a blood sample were recruited to the study as controls through the PHIs attached to the selected MOH areas in two districts. The estimated sample size for Kandy was 385 and that for Girandurukotte was 384.

### 2.3. Ethical Approval

Ethical approval for this study was obtained from the Institutional Ethical Review Committee, Faculty of Medicine, University of Peradeniya, Sri Lanka (2016/EC/64) and the Ethical Review Committee of the Graduate School of Medicine, Hokkaido University, Japan (M17-023).

### 2.4. Collection of Data and Blood Samples

Demographic data were collected using a structured questionnaire from both patients and controls. The content of the questionnaire as follows: (1) basic demographic data (age, gender, etc.), (2) family and personal past medical history, (3) occupational/agricultural involvement, and (4) environmental and animal exposure. Blood samples (5 mL from each participant at enrollment) were collected from January 2017 to February 2018 by well-trained medical laboratory technicians by venipuncture in accordance with all safety measures followed by informed consent which was provided in Appendix A.

### 2.5. Statistical Analysis

Data obtained from the structured questionnaires from both patients and controls, and the results of laboratory analysis of serum sample were entered into Microsoft Excel^®^. Demographic data were tabulated using Microsoft Excel^®^. All statistical analyses were conducted using Minitab software version 17. Exposure to hantavirus and possible risk factors for that exposure were evaluated using 2 × 2 contingency tables and Pearson chi-square (χ2) test. A p-value lower than 0.05 (*p* ≤ 0.05) was considered statistically significant. Fisher’s exact test was used when the counts were less than 5. Odds ratios (ORs) and 95% confidence intervals (CIs) were calculated in the group exposed to hantaviruses compared with the non-exposed group.

#### 2.5.1. Univariate Descriptive Analysis

Univariate descriptive analysis was performed to identify risk factors for exposure to hantaviruses in patients with renal disease and controls residing in Kandy and Girandurukotte.

#### 2.5.2. Comparison of Possible Risk Factors for Exposure to Hantavirus among Seropositive Individuals in the Two Areas

The possible risk factors for exposure to hantavirus were analyzed by 2 × 2 contingency tables to evaluate their association with exposure to hantavirus in patients with renal disease and controls residing in Kandy and Girandurukotte.

#### 2.5.3. Binary Logistic Regression Model

Binominal logistic regression models with exposure to hantavirus (seropositive) or no exposure to hantavirus (seronegative) as dichotomous response variables were used to identify variables or factors associated with the probability of an individual being infected. We tested whether the set of variables described (Appendix A) are able to predict the acquisition of infection. The responses were dichotomized before performing binary logistic regression analysis. We also used a logit link function to separately determine the influence of each of the independent variables. Thus, the analysis prevented independent variables from acting as confounding variables. The ORs were presented in favor of exposure to hantavirus in relation to the reference levels for binominal variables.

## 3. Results

### 3.1. Characteristics of Anti-Hantavirus Antibodies in Sera Obtained in Sri Lanka

#### 3.1.1. Reactivities in IFA

As shown in Appendix A, rNP-based antigen was successfully detected with anti-NP monoclonal antibodies but not with anti-GP antibodies. In the prototype THAIV antigen, NP and GP were detected as fine granular or fibrous structures in cytoplasm. Subsequently, rNP was observed as a defused pattern in the cytoplasm by anti-NP monoclonal antibody and no fluorescence pattern was observed by anti-GP monoclonal antibody in rNP-based antigen. A typical positive pattern by Kandy control serum #193 and a typical negative pattern by Kandy control serum #12 are shown in Appendix A. Therefore, samples that showed positive immunofluorescent patterns in both IFA tests. Prototype THAIV and rNP-based-IFA tests, were considered as positive.

#### 3.1.2. Reactivities in ELISA

In 2016, we reported a seropositivity rate of 54.5% in 132 CKDu patients living in Girandurukotte [7]. Serum sampled from these CKDu patients were examined by the aforementioned IFA tests using two antigens and were categorized as seropositive or seronegative. Sixty-one seropositive and 42 seronegative for IFA were subjected to ELISAs using both an HS103 antigen and an rNP-SEOV antigen (Appendix A). Differences were observed between the positive and negative groups in the two ELISAs. The HS103 antigen showed high OD values for the negative group, and establishment of a cut-off value was difficult. Therefore, ELISA using the HS103 antigen was not appropriate for serodiagnosis of CKDu patient sera. Even though ELISA using rNP-SEOV expressed in baculovirus vector showed higher specificity than that of HS103 ELISA, it was not considered as an appropriate method for serodiagnosis due to limitations on determining an optimum cut-off point. Therefore, IFA methods with two antigens were used for antibody detection in this study.

### 3.2. Demographic Characteristics and Serological Results of the Study Subjects

As shown in Table 1, there was a high seropositivity rate of 50% in CKDu patients and a seropositivity rate of 17.4% in healthy controls in Girandurukotte, similar to the rate in 2016. Strikingly, in Kandy, CKDu non-endemic areas also showed a positive rate as high as 18.9% in patients with renal disease, while a relatively low positive rate of 7.0% was found in healthy people. The results of univariate descriptive analysis for each group are shown in Appendix A.

### 3.3. Comparison of Possible Risk Factors for Exposure to Hantavirus

As shown in Table 2, in Kandy, several demographic characteristics were considered to be risk factors for exposure to the virus. There was a significant association between the age of 40 years or older and exposure to hantavirus infection (χ2 = 3.85, *p* = 0.05). Moreover, a larger percentage of males were exposed to the virus (χ2 = 6.79, *p* < 0.05). Paddy farming (χ2 = 11.08, *p* = 0.007) and engaging in agricultural activities (χ2 = 4.95, *p* = 0.030) were also identified as risk factors for exposure to hantaviruses, although storing crops at home was not found to be a risk factor. Seeing rodents (χ2 = 12.58, *p* < 0.001) in or around the house was a significant risk factor for exposure to hantaviruses among both renal disease patients and controls in Kandy. Complete data of this analysis is shown in Appendix A.

In Girandurukotte, age of 40 years or older (χ2 = 21.89, *p* < 0.001) and male gender (χ2 = 20.26, *p* < 0.001) were risk factors for exposure to hantaviruses (Table 2). In addition, paddy farming (χ2 = 18.30, *p* < 0.001) and storing crops at home (χ2 = 29.32, *p* < 0.001) were strongly associated with a higher frequency of exposure to hantavirus. As shown in Table 2, the presence of rodents inside houses or in surrounding areas (χ2 =38.42, *p* < 0.001) was significantly associated with hantavirus infections in the Girandurukotte area.

### 3.4. Binary Logistic Regression Models

Binary logistic regression models with exposure as the dichotomous response variable were used to identify statistically significant factors affecting exposure to hantavirus in the areas in this study (Table 3). The binary logistic regression model for Kandy showed that there was a statistically significant association between the response variable and the individual being a renal disease patient (OR: 2.64, 95% CI: 1.07 to 6.54, *p* < 0.05). Nevertheless, other interactions between independent variables, except the status of being a renal disease patient or a healthy control, were found to be nonsignificant. The logistic regression model for Girandurukotte showed that there was a strong association between exposure to hantavirus and renal disease in CKDu endemic areas (OR: 3.66, 95% CI: 2.01 to 6.64, *p* < 0.001). This finding indicated that the odds of an individual having been exposed to hantavirus was three-times higher for renal disease patients than for healthy controls in a CKDu endemic area. Moreover, there was a statistically significant association between seropositivity for hantaviruses and male gender with an OR of 2.79 (95% CI: 1.58 to 4.95, *p* < 0.001) in Girandurukotte. In addition, seroconverted individuals ≥40 years of age were found to be at a higher risk of exposure compared to individuals who were less than 40 years of age, although the difference was not significant (OR: 1.92, 95% CI: 0.96 to 3.82, *p* = 0.057).

## 4. Discussion

This is the first epidemiological study addressing hantavirus infection in renal disease patients in a CKDu endemic and a non-endemic area in Sri Lanka. A higher seroprevalence of hantavirus was found in kidney disease patients than in healthy controls in a CKDu endemic area, as reported in recent studies [7,24].

Girandurukotte, one of the CKDu hotspots, had a greater proportion of exposed individuals than Kandy. The study results provide interesting insights for the hypothesis of exposure to hantavirus as a possible risk factor for the development of CKDu in Sri Lanka [25]. Clinical records showed that 90 of the 104 Girandurukotte renal disease patients had diagnosed CKDu cases according to the 2009 guideline [22]. Significant associations were found between renal disease and hantavirus exposure in both Girandurukotte and Kandy. The majority of seropositive renal disease patients in Girandurukotte were CKDu patients, but none of the renal disease patients in Kandy were diagnosed with CKDu. Furthermore, the nine seropositive renal disease patients in Kandy included seven CKD patients, one end-stage renal disease patient, and one acute kidney injury (AKI) patient. The diagnosis of CKDu in Sri Lanka is made according to guidelines that are solely based on clinical parameters (GFR levels). These results confirm the existence of sporadic hantavirus infection even in Kandy after the first report by Gamage et al. in 2011 [12].

In Girandurukotte, the majority of residents are engaged in agriculture such as paddy farming. In contrast, fewer people are engaged in paddy farming in Kandy than in Girandurukotte. In this study, we showed that age of 40 years or older, male gender, paddy farming, storing crops at home, and witnessing rodents in or surroundings of the house were high-risk factors for hantavirus infection in CKDu endemic areas. Moreover, the presence of anti-hantavirus antibodies was the most significant risk factor among renal disease patients. In Kandy, individuals in the exposed group were older than those in the non-exposure group and were more likely to be male, paddy farmers, and to have sighted rats. However, storing crops at home was not found to be a risk factor for exposure to hantavirus in Kandy. This finding suggested that hantavirus infection in this area might not cause by rodents invading the house and suggests the rationale of lower seroprevalence in females than in males.

Renal disease patients in both areas tended to be older than the controls and there was a larger percentage of males than that in the control groups. However, our results were likely to have been affected by biased information provided by the participants with regard to interactions with rodents and their excreta during the survey-based interview. Despite these constraints, this study provides empirical information of exposure to hantavirus as a risk factor for renal disease in a CKDu endemic area.

In 2016, we reported high seropositivities in the CKDu patients living in Girandurukotte [7]. By using their sera, we established a reliable serodiagnostic method for this study. These sera were also applied to ICG by using HS103 antigen as mentioned in a previous report [26]. They showed varied reactivities. Some showed relatively strong bands, whereas others showed faint or no bands (data not shown). Although ICG tests are considered to be the best method for field surveys, they require high affinity and avidity of antigen-antibody binding. In order to develop ICG method specific to Sri Lanka, it is necessary to use an antigen derived from an orthohantavirus specific to Sri Lanka.

In Kandy, which is considered to be a non-endemic area, seroprevalences in renal disease patients and controls (18.0% and 7.0%, respectively) were lower than those in Girandurukotte. Furthermore, in this study, the rate of seroprevalence for hantavirus in the apparently healthy population was higher than that reported elsewhere. The seroprevalence of THAIV among healthy controls in Thailand has not been reported yet. However, seroprevalence of 0.4% (1/260) for THAIV was found in febrile patients in Surin Province of Thailand who were suspected of having leptospirosis and were serologically negative for Leptospira antigens [10]. In Chile, the seroprevalences for Andes orthohantavirus in apparently healthy individuals were reported to be 1.3% and 1.5% in rural and urban poor communities, respectively [27]. Similarly, a cross-sectional survey was carried out in an endemic area of the state of Minas Gerais, a peri-urban and rural area in Brazil, to assess the proportion of persons exposed to hantaviruses. Antibody positivity was determined to be 3% by ELISA using a recombinant antigen of Araraquara orthohantavirus associated with HPS [28], whereas the results of the current study showed a seropositive rate of 17.4% in controls living in rural agricultural communities of Girandurukotte and a seropositive rate of 7% in controls living in peri-urban Kandy.

The recent study showed that the innate immune response was activated in CKDu patients, including interferon, inflammasome and triggering receptor expressed in myeloid cell-1 (TREM1) signaling. Transcriptome analysis suggests that viral infections and fluoride or other toxic environmental factors appear to be contributing to the molecular mechanisms underlying the development of CKDu in Sri Lanka [29,30,31]. However, the causes of CKDu have remained unknown for more than 20 years. The results of this study showed that exposure to hantavirus infection is a possible risk factor underlying renal disease in both endemic and non-endemic areas. Antibodies detected in these individuals were associated with a lack of past clinical symptoms or a medical history of being exposed to hantaviruses, suggesting that infection with THAIV-related hantavirus might be an asymptomatic or a mild infection with nonspecific clinical symptoms at the onset of the disease.

Several studies carried out in Europe has revealed pathogenesis related to mild to moderate forms of kidney damage during several hantavirus infections. In Northern Germany, Dobrava-Belgrade virus (DOBV) infected patients showed a diverse range of mild to moderate clinical manifestations associated with kidney injury [32]. In another study, patients with nephropathia epidemica (NE) often referred as a mild form HFRS caused by Puumala virus exhibited upregulation of biomarkers of inflammation suggested damage in kidney proximal tubule and leukocyte chemotaxis at the onset of NE in a cohort of patients in a Russian hospital [33]. On the other hand, CKDu in Sri Lanka is known to develop from tubular damage [34,35]. Several Old World hantaviruses have been reported to cause AKI but not HFRS [36]. Furthermore, it has been reported that chronic proteinuria continues even several years after recovery from hantavirus infection [32,37]. Thus, there are similarities between CKDu in Sri Lanka and hantavirus infection. In addition to the epidemiological data obtained in this study, it is necessary to clarify the pathological association between CKDu and hantavirus infection.

Thus, there is a necessity for island-wide sero-epidemiological studies on exposure to hantavirus using both retrospective (renal disease patients) and prospective (nonspecific febrile patients) approaches. Future directions of this study are focused on identifying the reservoir host carrying the THAIV-related hantavirus in these CKDu endemic areas and understanding the dynamics of its transmission to humans. The pathological and molecular biological mechanisms underlying the progression of renal damage due to circulating THAIV-related hantaviruses should also be investigated to understand the disease dynamics.

## 5. Conclusions

The results of this study showing a very high seroprevalence of hantavirus in renal disease patients in the studied area that is endemic for CKDu provide an insight into the crucial role of hantavirus in renal disease. Since very few investigators have identified CKDu as a disease with an infectious etiology, communication among research groups worldwide are important to develop internationally accepted definitions for CKDu in different geographical areas, which will ultimately aid in developing effective means of preventing the disease.

## Figures and Tables

**Figure 1 viruses-11-00700-f001:**
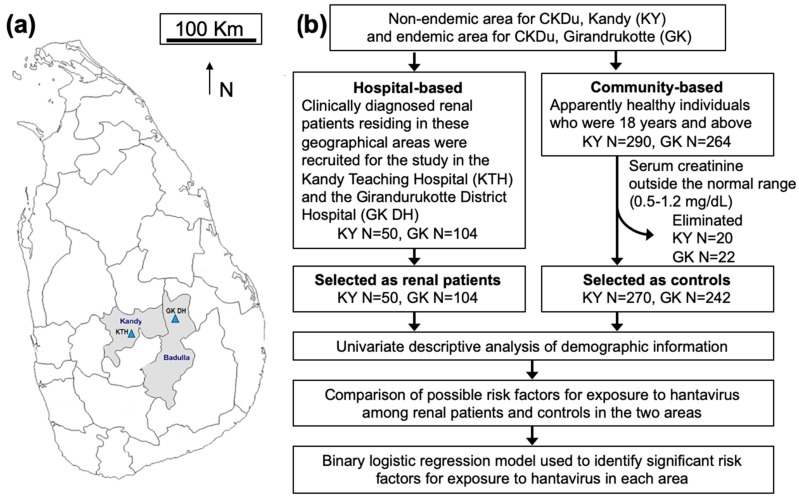
Study area and design. (**a**) Kandy and Badulla districts of Sri Lanka and the sampling locations. The map shows the locations of Kandy Teaching Hospital (KTH) and Girandurukotte District Hospital (GK DH). (**b**) Flow chart summarizing the selection process for renal patients and controls and the statistical analyses. Blood and demographic information were collected from individuals. The presence of serum IgG antibodies against hantaviruses was measured and statistically analyzed.

**Table 1 viruses-11-00700-t001:** Demographic characteristics of the study subjects.

Area	Status	Gender	Number	Age Mean (SD)	Serum Antibody Positive to Hantavirus (%)	Occupation Paddy Farming
Girandurukotte	Renal patients	Male	70	58 (12.3)	42 (60.0%)	65
		Female	34	60 (10.2)	10 (29.4%)	27
		Total	104	59 (11.7)	52 (50.0%)	92
	Community	Male	98	47 (14.8)	25 (25.5%)	68
		Female	144	45 (14.0)	17 (11.8%)	44
		Total	242	46 (14.3)	42 (17.4%)	112
Kandy	Renal patients	Male	28	54 (12.6)	7 (25.0%)	2
		Female	22	54 (11.1)	2 (9.1%)	1
		Total	50	54 (11.8)	9 (18.0%)	3
	Community	Male	126	48 (14.5)	10 (7.9%)	10
		Female	144	46 (13.5)	9 (6.3%)	3
		Total	270	47 (14.0)	19 (7.0%)	13

**Table 2 viruses-11-00700-t002:** Comparison of possible risk factors for exposure to hantavirus in Kandy and Girandurukotte.

	Kandy				Girandurukotte			
Category	Anti-Hantavirus Antibody		χ2	*p*-Value	Anti-Hantavirus Antibody		χ2	*p*-Value
	Positive	Negative			Positive	Negative		
**Age**								
Age category of ≥ 40 years								
Renal patients	8	38	3.85	0.050	48	47	21.89	<0.001
Control	14	166			32	116		
Age category of < 40 years								
Renal patients	1	3	2.42	0.235	4	5	7.99	0.005
Control	5	85			10	84		
**Gender**								
Female								
Renal patients	2	20	0.25	0.642	10	24	6.53	0.011
Community	9	135			17	126		
Male								
Renal patients	7	21	6.79	0.010	42	28	20.3	<0.001
Community	10	116			25	73		
**Occupation**								
Paddy farming								
Renal patients	3	0	11.08	0.007	47	45	18.3	<0.001
Community	1	12			25	87		
Other occupations								
Renal patients	6	41	1.81	0.178	5	7	18.3	0.009
Community	18	239			17	113		
**Storing crop at house**								
Yes								
Renal patients	2	4	1.44	0.260	48	49	29.32	<0.001
Community	4	26			28	131		
No								
Renal patients	7	37	4.85	0.028	4	3	6.55	0.027
Community	15	225			14	69		
**Rats seen at home or surroundings**								
Yes								
Renal patients	8	25	12.58	<0.001	45	39	38.42	<0.001
Community	12	194			34	163		
No								
Renal patients	1	16	0.39	0.535	7	13	2.31	0.128
Community	7	57			8	37		

**Table 3 viruses-11-00700-t003:** Results of the binary logistic regression models for Kandy and Girandurukotte for prediction of the risk of an individual being seropositive for hantavirus.

Predicted Risk Factors	Kandy			Girandurukotte		
	χ2	*p*-Value	OR (95% CI)	χ2	*p*-Value	OR (95% CI)
Status						
Renal patient	4.08	0.044	2.64 (1.07 to 6.54)	18.83	<0.001	3.66 (2.01 to 6.64)
Age						
Age ≥ 40 years	0.01	0.935	1.04 (0.38 to 2.86)	3.62	0.057	1.92 (0.96 to 3.82)
Gender						
Male	0.70	0.402	1.43 (0.62 to 3.31)	12.88	<0.001	2.79 (1.58 to 4.95)
Occupation						
Farmer	1.02	0.313	2.18 (0.49 to 9.63)	0	0.985	0.99 (0.45 to 2.18)
Engaged in agriculture-related activities						
Yes	2.76	0.097	2.53 (0.88 to 7.27)	1.41	0.235	1.82 (0.67 to 4.91)
Store crops at house						
Yes	0.02	0.897	0.92 (0.24 to 3.47)	1.30	0.255	0.65 (0.32 to 1.36)
Sighting rodents at home or surrounding						
Yes	0.06	0.446	0.67 (0.25 to 1.85)	2.21	0.137	1.86 (0.81 to 4.24)
Presence of rodent excreta at home or surrounding						
Yes	0	0.982	1.01 (0.40 to 2.54)	1.72	0.19	0.65 (0.34 to 1.24)

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
