# Peer review of "Exposure to Hantavirus is a Risk Factor Associated with Kidney Diseases in Sri Lanka: A Cross Sectional Study"

_viruses, 2019, doi:10.3390/v11080700_

Round 1

Reviewer 1 Report

Major points:

1.       Although data on IFA is convincing, the statement that IFA is better than ELISA for detection of antibodies seems to be far stretched. It is correct, ELISA requires sophisticated equipment and personnel training, IFA has multiple limitations as well. Therefore, statement that IFA is better for antibody detection as compared to ELISA should be revised.

2.       Sections 311 and 312 should be removed to supplemental data 

3.       Sections 331 and 332 could be combined as results are similar

4.        Papers addressing kidney pathogenesis should be cited: doi: 10.1155/2018/8658507;  doi.org/10.1159/000486322; dx.doi.org/10.1136/bcr-2014-205529

Minor points:

English requires extensive editing.

Author Response

Thank you for your valuable comments to improve our manuscript. We have described the following for your comments

Major Points: 

1.     Revised accordingly fromlines (229-236):“Differences were observed between the positive and negative groups in both ELISAs. The HS103 antigen showed high OD values for the negative group, and establishment of a cut-off value was difficult. Therefore, ELISA using the HS103 antigen was not appropriate for serodiagnosis of CKDu patient sera. Even though ELISA using rNP-SEOV expressed in baculovirus vector showed higher specificity than that of HS103 ELISA, it was not considered to be an appropriate method for serodiagnosis due to limitations on determining an optimum cut-off point. Therefore, the use of IFA methods with two antigens was used for antibody detection in this study.”

2.     According to the reviewer’s comment, we have moved sections 3.1.1. and 3.1.2 to Supplementary data 6 and 7.

3.     According to the reviewer’s comment, we have combined results of the sections 3.3.1 and 3.3.2.

4.     Revised accordingly citing four of the articles as suggested from lines (352-364) in the Discussion section; Several studies carried out in the Europe has revealed pathogenesis related to mild to moderate forms of kidney damage during several hantavirus infections. In Northern Germany, Dobrava-Belgrade virus (DOBV) infected patients showed diverse range of mild to moderate clinical manifestations associated with kidney injury [32]. In another study, patients with nephropathia epidemica (NE) often referred as a mild form HFRS caused by Puumala virus exhibited upregulation of biomarkers of inflammation suggested damage in kidney proximal tubule and leukocyte chemotaxis at the onset of NE in a cohort of patients in a Russian hospital [33]. On the other hand, CKDu in Sri Lanka is known to develop from tubular damage [34, 35]. Several Old World hantaviruses have been reported to cause AKI but not HFRS[36]. Furthermore, it has been reported that chronic proteinuria continues even several years after recovery from hantavirus infection [32, 37]. Thus, there are similaritiesbetween CKDu in Sri Lanka and hantavirus infection. In addition to the epidemiological data obtained in this study, it is necessary to clarify the pathological association between CKDu and hantavirus infection.”

Minor Points: 

English editing was done according to reviewer’s comment.

We believe that the manuscript has been greatly improved and hope it has reached your journal’s standards. Once again, we acknowledge and very much appreciate your comments and suggestions, which were immensely helped in improving the quality of our manuscript.

Thanking you,

Sincerely yours,

Dr. Kumiko Yoshimatsu, DVM, PhD

Reviewer 2 Report

In the study, it was observed that probably asymptomatic or mild infection of Thailand orthohantavirus (THAIV) is a risk factor for chronic kidney disease of unknown etiology (CKDu). The study is well-performed and the text is well-written. Th results are interesting and important. I have only two minor comments.

Because I am a clinician (internist and nephrologist), I consulted Professor Antti Vaheri (University of Helsinki, Finland). He told that the serological diagnostics that is an important part of the study, has been adequately performed.

What was the patient group defined as “Clinically diagnosed renal patients”? Did all patients have reduced glomerular filtration rate (GFR)? Or did some of them have only proteinuria and/or hematuria?

Author Response

We are thankful for your positive and helpful suggestions on our manuscript. 

Regarding the definition of “Clinically diagnosed renal patients” we had to follow on several guidelines.

Patients we recruited from the Nephrology and Transplantation Unit of Kandy Teaching Hospital (KTH) were CKD patients defined according to the National Kidney Foundation (NKF) based on the National Kidney Disease Outcomes Quality Initiative (KDOQI) criteria guidelines where the standard CKD classification (stage I–V) are represented based on the GFR level of kidney function (National Kidney Foundation (2002), KDOQI Clinical Practice Guidelines for Chronic Kidney Disease: Evaluation, Classification, and Stratification). 

When we first started this study in early 2016, we followed the clinical diagnosis criteria for CKDu patients in Sri Lanka that was established by the World Health Organization (WHO) and the Ministry of Health of Sri Lanka in 2009 (Circular No. Epid/392/2008/25 dated February 10th, 2009). The CKDu cases definitions were modified from KDOQI as mentioned in detailed below; 

1. No past history of or current treatment for diabetes mellitus or chronic and/or severe hypertension, snake bite, urological disease of known aetiology or glomerulonephritis

2. Normal HbA1C (< 6.5%)

3. BP < 160/100 mmHg untreated or < 140/90 on up to two antihypertensive agents

However, the new case definition of CKDu has been described in 3 levels as suspected, probable and confirmed cases according to the guidelines by together with the Presidential Task Force on CKDu in October 2016. It is listed as below; 

·      A suspected CKDu case is defined when the eGFR < 60 mL/min using CKD EPI equation (Levey et al., 2009), one-time measurement using standardized methods for creatinine measurement OR albuminuria > = 30 mg/g. 

·      Probable CKDu cases were defined if the repeat assessment after 12 weeks of eGFR <60 mL/min using CKD EPI equation OR repeat albuminuria > = 30 mg/g.

·      Confirmed CKDu cases consist with above mentioned criteria for probable CKDu and in addition including histo-pathological features consistent with CKDu on renal biopsy. 

Exclusion criteria for these case definitions are as follows; past history or current treatment for diabetes mellitus or on fasting plasma glucose >126 mg/dL, hypertension on treatment with more than two drugs or untreated blood pressure of more than 160/100 mmHg and Haematuria of >10 red blood cells/HPF. Furthermore, presence of polycystic kidney, congenital malformation, obstructive nephropathy by ultrasound imaging or other known causes of CKD such as autoimmune diseases, glomerular diseases, kidney stones or any other obstruction in the urinary tract based on the clinical evaluation and laboratory investigations.

Regardless of the clinical definitions, in the actual hospital set-ups we referred the information on patient’s clinical record books as diagnosed by the Consultant Nephrologist (Dr. NN) at KTH and residential clinician (Dr. LN) at Girandurukotte District hospital (GK DH) and we recruited patients who regularly visit the renal clinics at the two hospitals. In GK DH > 85% of the renal patients were CKDu according to the 2009 guidelines but very few biopsy proven cases due to poor facilities in the local settings. 

Manuscript was revised accordingly (147-150):“Renal patients were recruited according to the information on patient’s clinical record books as diagnosed by the Consultant Nephrologist at Nephrology and Transplantation Unit of Kandy Teaching Hospital (KTH) and residential clinician at the renal clinic, Girandurukotte District hospital (GK DH) according to the standard guidelines [22, 23].”

We believe that the manuscript has been greatly improved and hope it has reached your journal’s standards. Once again, we acknowledge and very much appreciate your comments and suggestions, which were immensely helped in improving the quality of our manuscript.

Sincerely yours,

Dr. Kumiko Yoshimatsu, DVM, PhD

Round 2

Reviewer 1 Report

Revised manuscript can be accepted for publication